# Is There a Role of Warburg Effect in Prostate Cancer Aggressiveness? Analysis of Expression of Enzymes of Lipidic Metabolism by Immunohistochemistry in Prostate Cancer Patients (DIAMOND Study)

**DOI:** 10.3390/cancers15030948

**Published:** 2023-02-02

**Authors:** Giorgio Ivan Russo, Maria Giovanna Asmundo, Arturo Lo Giudice, Giuseppe Trefiletti, Sebastiano Cimino, Matteo Ferro, Riccardo Lombardo, Cosimo De Nunzio, Giuseppe Morgia, Eliana Piombino, Maria Failla, Rosario Caltabiano, Giuseppe Broggi

**Affiliations:** 1Urology Section, Department of Surgery, University of Catania, 95124 Catania, Italy; 2Department of Urology, European Institute of Oncology, IRCCS, 20139 Milan, Italy; 3Department of Urology, “Sant’Andrea” Hospital, “La Sapienza” University, 00185 Rome, Italy; 4Department of Experimental Oncology, Mediterranean Institute of Oncology (IOM), 95029 Catania, Italy; 5Pathology Unit, Garibaldi Hospital, 95123 Catania, Italy; 6Department of Medical and Surgical Sciences and Advanced Technologies “G. F. Ingrassia”, Anatomic Pathology, University of Catania, 95124 Catania, Italy

**Keywords:** prostate cancer, Warburg effect, prognosis, radical prostatectomy, lipidic, metabolism, mortality

## Abstract

**Simple Summary:**

Prostate Cancer (PCa) is still ranked as the first cancer in the male population and evidence has suggested an alteration of glycemic and lipidic metabolism that are related to its progression and prognosis. We demonstrated that the expression of ATP-lyase, CPT-1a, SCD, SREBP, ACC-1, and FAS were associated with AR. Finally, SCD+ expression in PCa patients with total cholesterol ≥ 200 mg/dL was independently associated with ISUP ≥ 4, and CPT-1a+ was associated with biochemical recurrence. Our results support the evidence that the manipulation of lipidic metabolism could serve in the future to contrast PCa progression.

**Abstract:**

Prostate Cancer (PCa) is still ranked as the first cancer in the male population and evidences have suggested an alteration of glycemic and lipidic metabolism that are related to its progression and prognosis. The aim of the study is to investigate associations between enzymes’ expression, especially involved in the lipidic pathway, and PCa aggressiveness. We retrospectively analyzed data from 390 patients with PCa or benign prostatic hyperplasia (BPH) at the Department of Urology, University of Catania. Immunohistochemical slides were evaluated for the expression of proteins related to glucose and lipidic metabolism. A total of 286 were affected by PCa while 104 by BPH. We demonstrated that ATP-lyase (odds ratio [OR]: 1.71; *p* < 0.01), fatty acid synthase (OR: 4.82; *p* < 0.01), carnitine palmitoyl transferase-1a (OR: 2.27; *p* < 0.05) were associated with androgen receptor (AR) expression. We found that steaoryl Co-A desaturase expression in PCa patients with total cholesterol ≥ 200 mg/dL was independently associated with ISUP ≥4 (OR: 4.22; *p* = 0.049). We found that CPT-1a+ was associated with biochemical recurrence (hazard ratio: 1.94; *p* = 0.03]). Our results support the evidence that the manipulation of lipidic metabolism could serve in the future to contrast PCa progression.

## 1. Introduction

Prostate Cancer is still ranked as the first cancer diagnosed in the male population in 2022 accounting alone for more than 27% of incident cases in men [1]. It is well known that a family history of PCa is one of the strongest risk factors for the development of the pathology; in fact, the percentage of PCa related to hereditary components accounts for 5–15% of cases [2], but also de-novo mutations have a considerable role [3]. In recent years, a notable interest has arisen regarding the impact of modifiable risk factors in the development and progression of PCa. Current evidence suggests that conditions such as metabolic syndrome (MetS), hypertension, diabetes mellitus, obesity, and cigarette smoking may be implied into tumor cell mutation, disease promotion, and high-grade tumor incidence [4]. MetS is a cluster of cardiometabolic risk factors characterized by high blood pressure, abdominal obesity, low level of HDL, hyperglycemia, and hypertriglyceridemia [5]; moreover, this condition is commonly combined with increased circulating levels of inflammatory mediator and growth factors that may contribute to the PCa carcinogenesis process [4,6]. All these conditions share an altered metabolic control system and abnormal nutrient-sources management; it results in an anomalous environment and exceeding nutrients that promote aberrant cell proliferation [7].

Interestingly, the so-called Warburg effect claims that even when oxygen is available, tumor cells tend to degrade it via anaerobic glycolysis [8]. Even though the non-oxidative pathways may appear to be an inefficient use of resources, it supports quick cell division through faster biomass increase [9]. All of this evidence supports the theory that metabolic enzymes can clearly take part in tumor promotion and development. For these reasons, increasing attention has been paid to the pivotal role of enzymes involved in energetic pathways, especially in the lipidic one. De novo fatty acid synthesis is quite deficient in almost every tissue except for some specific organs; however, studies highlighted that β-oxidation of fatty acid is the main bioenergetic pathway in PCa [10]. Indeed, an upregulation of lipogenic and lipolytic enzymes and their mRNA had been reported in PCa [11]. Acetyl-CoA, the essential building block for fatty acid metabolism, plays crucial roles at the interface of metabolism, signaling, and gene regulation [12,13].

Inhibiting lipogenic enzymes such as fatty acid synthase (FASN), acetyl-CoA carboxylase (ACC), or ACLY produces anti-cancer effects both in prostate cancer cell lines and mouse models [14,15].

Singh et al. evidenced that administration of sulforaphane (SFN) in preclinical mouse model results in dose-dependent downregulation of Acetyl-CoA Carboxylase 1 (ACC1) and Fatty Acid synthase (FASN) in human prostate cancer cell line; moreover, a decreased expression of these enzymes seems to be related to in vivo and in vitro PCa cell growth suppression. Additionally, a not-significant decreasing of Sterol Regulatory Element Binding Protein (SREBP-1), Carnitine Palmitoyl transferase 1 (CPT1A), Acetyl Carboxylase (ACAC-2) and ATP citrate lyase has been registered, too [16].

In this regard, previous studies have highlighted that prostate-specific membrane antigen (PSMA), Stearoyl-CoA Desaturase 1 (SCD1) and Insulin-like Growth factor 1 Receptor (IGFR1) have an altered expression in PCa [17,18].

Thus, based on these premises, the aim of the study is to investigate associations between enzymes’ expression, especially those involved in the lipidic pathway, and PCa aggressiveness.

## 2. Materials and Methods

In the present study, we retrospectively analyzed data from 390 patients who underwent Radical Prostatectomy, if affected by PCa, or Trans Urethral Resection of Prostate (TURP), if diagnosed with Benign Prostatic Hyperplasia (BPH), between 2010 and 2020 at the Department of Urology, University of Catania.

### 2.1. Immunohistochemistry (IHC)

Immunohistochemical slides were evaluated by three pathologists (G.B., E.P. and R.C.) with no information on patient clinical data, as previously described [19,20]. Immunohistochemical analyses were performed according to previous methodology [19], As concerning new experiments, we used FASN (fatty acid synthase) antibody (C20G5; Rabbit IgG monoclonal, 1:50–1:200 dilution) [21], Carnitine palmitoyltransferase I (CPT-1) (15184-1-AP, Proteintech, Rosemont, IL, USA, 1:1500 dilution, Benchmark XT I-VIEW DAB detection kit from Ventana, Roche Group, Basel, Switzerland) [22], Sterol regulatory element-binding protein (SREBP1) (ab28481, rabbit polyclonal antibody, 1:50–1:500 dilution, Abcam, Cambridge, UK) [23], ATP citrate lyase: anti-ACLY (ab40793, rabbit monoclonal antibody, Abcam) [23], Stearoyl-CoA desaturase-1 (SCD) (BS-3787R, rabbit polyclonal, 1:400 dilution, Bioscience, Allentown, PA, USA) [24], Acetyl-CoA Carboxylase 1 (ACC-1) (cat. #4190, rabbit monoclonal, 1:50 dilution, Cell Signaling Technology, Danvers, MA, USA) [25]. The scoring applied was conducted following previous methodology [19,20,26].

### 2.2. Statistical Analysis

Continuous variables are presented as the median and interquartile range (IQR) and were tested by the Mann–Whitney U test due to their not-normal distribution. A chi-square test was applied for categorical variables. Univariate and multivariate logistic regression has been used to test independent variables associated with IHC scores. PC was classified into low, intermediate, and high according to EAU guidelines [27]. Biochemical recurrence was tested by applying the Kaplan–Meier curve. A significance of *p* < 0.05 was considered to show differences between the groups. Data analysis was performed under the guidance of our statistics expert, using StataCorp. 2021 (Stata Statistical Software: Release 17).

## 3. Results

A total of 286 were affected by PCa while 104 by BPH. Median age (years) was 69.0 (interquartile range [IQR]: 64.0–74.0), median PSA was 6.5 (IQR: 4.27–10.0), median cholesterol was 185.5 (IQR: 158.0–214.0), median triglycerides was 100.0 (IQR: 69.0–143.5) and median fasting blood glucose was 97.0 (IQR: 88.0–110.0). We found great levels of total cholesterol (mg/dL) in PCa vs. BPH (191.5 [IQR 168–221.5] vs. 175.0 [IQR 150.0–199.0], *p* < 0.01) while less level of Triglycerides (mg/dL) (97 [IQR 68.0–130.0] vs. 117 [IQR 69.0–164.0], *p* < 0.01). The number of diabetes patients was 44 (15.38%) in PCa and 28 (26.92%) in BPH (*p* < 0.01). Total case of BCR was 49 (17.13%). These results were adapted from [19].

At the IHC analysis we found a median IRS of 2.0 (IQR: 0.0–2.0) for PSMA, of 8.0 (IQR: 4.0–9.0) for IR-α positive score, of 0.0 (IQR: 0.0–2.0) for IR-β positive score, of 1.0 (IQR: 0.0–4.0) for IGF-1R positive score, of 4.0 (IQR: 0.0–4.0) for AR, of 4.0 (IQR: 4.0–8.0) for SRSF-1 positive score, of 4.0 (IQR: 3.0–8.0) for ACC-1, of 4.0 (IQR: 3.0–8.0) for ATP citrate lyase, of 1.0 (IQR: 0.0–4.0) for CPT-1a, of 2.0 for (IQR: 0.0–4.0) for SCD-1, of 3.0 (IQR: 1.0–4.0) for SREBP-1 and of 6.0 (IQR: 2.0–9.0) for FAS.

Table 1 lists the ATP citrate lyase expression according to the IHC score and its relationship with clinical and pathological data.

In particular, we found that patients with diabetes had a lower rate of H-IRS respect those without (15.38% vs. 26.92%; *p* < 0.01). We also observed a greater rate of H-IRS for ATP citrate lyase in patients with high Ki-67 (*p* = 0.02), AR (*p* < 0.01), PSMA (*p* < 0.01), IR-α (*p* < 0.01), IGF-1R (*p* < 0.01), SRSF-1 (*p* < 0.01), SCD (*p* < 0.01), SREBP-1 (*p* < 0.01), FAS (*p* < 0.01) and ACC-1 (*p* < 0.01).

We also evaluated the relationship between Carnitine palmitoyltransferase-1a in our cohort and we found a high rate of H-IRS in patients with high AR (*p* < 0.01), IR-α (*p* < 0.01), IR-β (*p* = 0.02), SRSF-1 (*p* < 0.01), SREBP-1 (*p* = 0.01), FAS (*p* < 0.01) and ACC-1 (*p* < 0.01) (Table 2).

Table 3 shows the univariate logistic regression analysis combining all IHC results with clinical and pathological variables. We demonstrated that ATP-lyase high-IRS was associated with high-IRS of AR (OR: 1.71; *p* < 0.01) and FAS (OR: 4.82; *p* < 0.01), CPT-1a high-IRS was associated with AR (OR: 2.27; *p* < 0.05); IR-alpha (OR: 2.55; *p* < 0.01), IR-beta (2.45; *p* < 0.01), SCD (OR: 2.15; *p* < 0.01); SREBP (OR: 2.95; *p* < 0.01) and FAS (OR: 2.16; *p* < 0.01). Furthermore, ACC-1 high IRS was an independent predictor of high IRS of AR (OR: 3.65; *p* < 0.01), PSMA (OR: 1.80; *p* < 0.01); IR-alpha (OR: 9.99; *p* < 0.01), SCD (OR: 2.63; *p* < 0.01); SREBP (OR: 2.53; *p* < 0.01) and with FAS (OR: 11.29; *p* < 0.01). 

Interestingly, we found that SCD+ expression in PCa patients with total cholesterol ≥ 200 mg/dL was independently associated with ISUP ≥4 (odds ratio [OR]: 4.22 [95% CI 1.01–17.95); *p* = 0.049).

We performed the multivariate logistic regression analysis and we found that PSA (OR: 1.33; *p* < 0.01), ATPLy (OR: 3.16; *p* = 0.049), FAS (OR: 6.58; *p* < 0.01) and PSMA (OR: 8.05; *p* < 0.01) were independent predictors of PCa (the model was adjusted for age, total cholesterol and triglycerides).

After a median follow-up of 32 months, we observed 49 (17.13%) BCR. At the univariate Cox regression analysis, we found that CPT-1a+ was associated with BCR (HR: 1.94 [95% CI 1.05–3.59]; *p* = 0.03]) (Figure 1).

## 4. Discussion

In the present study, we demonstrated that in patients with PCa, there is a significant impact of lipidic metabolism on the prognosis and aggressiveness. In particular, specific enzymes such as ATP-lyase, FAS, CPT-1a, and ACC-1 were associated with AR expression and in particular, SCD+ expression was an independent predictor of PCa aggressiveness (ISUP ≥ 4).

Our study offers new insights in the terms of a better understanding of metabolism in PCa and even further a better elucidation of the potential impact of the Warburg effect.

The Warburg effect is a mechanism that occurs in cancer in which tumor cells tend to “ferment” glucose into lactate even in the presence of sufficient oxygen to support mitochondrial oxidative phosphorylation [9]. This led to the production of large amounts of lactate regardless of the availability of oxygen in a modified metabolism called “aerobic glycolysis”.

This enhanced glucose catabolism results in an excess of the glycolytic end product pyruvate [28]. Furthermore, the excess in pyruvate enters the mitochondrial matrix and it is converted into acetyl CoA [28]. Consequently, at this level, there is an increased activity of citrate synthase that catalyzes the condensation of acetyl CoA with oxaloacetate by producing citrate [28]. This product is fundamental since it is exported to the cytosol in proliferating cells and used as a biosynthetic precursor for lipogenic pathways. ATP-citrate lyase (ACLY) is a cytosolic enzyme that is fundamental to generating acetyl CoA from citrate [29] and is a precursor of fatty acids. ACLY has been associated with cancer and studies have demonstrated that some types of tumor cells can be suppressed by its inhibition [30,31,32].

In this context, it has been reported that PCa cells produce fatty acids for their energy through a de novo synthesis [33]. This mechanism of fatty acid production seems crucial for its progression [33]. Previous studies have shown that PCa cells overexpress certain markers that are key in the ability to produce de novo lipids [34] such as fatty acid synthase (FASN), sterol regulatory element binding protein 1 (SREBP1), and steroyl CoA desaturase among others [34].

Further studies demonstrated that sterol O-acyltransferase 1 (SOAT1) promotes liposynthesis and consequent PCa proliferation by its action on Stearoyl-CoA Desaturase 1 (SCD1). Moreover, enzymes’ expression seems to be strictly related to clinical stage, tumor grading, Gleason Score, and presence of lymphnode metastasis in PCa patients [18].

PCa may also overexpress the Sterol O-acyltransferase 1 (SOAT1) [18]. Liu et al. found that the expression of SOAT1 was elevated in PCa tissues and it was associated with lymph node metastasis (*p* = 0.006), clinical stage (*p* = 0.032), grading (*p* = 0.036), and Gleason score (*p* = 0.030). Moreover, SOAT1 targeted Stearoyl-CoA Desaturase 1 (SCD1) promoting the proliferation and liposynthesis of cells. Finally, SOAT1 contributed to the progression of PCa via the SREBF1 pathway [18].

Pharmacological or gene therapy aims to reduce the activity of enzymes involved in the de novo synthesis of fatty acids, FASN, ACLY (ATP citrate lyase), or SCD-1 (Stearoyl-CoA Desaturase) in particular, that may result in cells growth arrest [35]. Interestingly, castration-resistant PCa cells exhibit increased de-novo lipid synthesis compared to hormone-sensitive PCa cells and enzalutamide-resistant cells [36]. To reverse the increase of de-novo lipid synthesis and prevent enzalutamide resistance, authors demonstrated that the combination of SCD-1 inhibitors and enzalutamide considerably inhibits the growth of PCa xenografts [36].

Another crucial step is the activation of the Carnitine palmitoyltransferase 1 catalyzes the rate-limiting step of fatty acid oxidation [37]. Abudurexiti et al., using data from The Cancer Genome Atlas and Gene Expression Omnibus databases, demonstrated that CPT-1b expression was associated with was significantly associated with worse disease-free survival and overall survival and using in-vitro models that AR may regulate CPT1B expression and activity via specific binding site [37].

Schlaepfer et al. interestingly reported that using the combination of Etomoxir and Orlistat resulted in a synergistic decreased viability in LNCaP, VCaP and patient-derived benign and PCa cells and also AR downregulation [38].

Moving into the conclusion, the importance of a better understanding of the relationship between lipid metabolism and PCa progression is crucial and it should be considered in the near future in order to improve survival in PCa patients. Moreover, our results help in identifying specific metabolic markers for PCa diagnosis using innovative technologies [39].

Finally, we would like to address some limitations. Firstly, we were not able to investigate the relationship between body weight composition and IHC lipidic expression. Secondly, we did not investigate the relationship between metabolism and potential genomic alterations. Thirdly, the short follow-up in our cohort was not considerably sufficient to address overall survival. Finally, we did not take into account that cribriform histology which is gaining much more interest as an independent worse prognostic factor [40].

On the other hand, our study represents one of the few that reported tissue alterations of lipidic metabolism in PCa and their association with progression and prognosis.

## 5. Conclusions

In this study, we reported immunohistochemistry expression of proteins related to lipidic metabolism and their relationship with PCa prognosis and progression. In particular, we demonstrated that the expression of ATP-lyase, CPT-1a, SCD, SREBP, ACC-1, and FAS were associated with AR. Finally, SCD+ expression in PCa patients with total cholesterol ≥ 200 mg/dL was independently associated with ISUP ≥4 and that CPT-1a+ was associated with biochemical recurrence. Our results support the evidences that the manipulation of lipidic metabolism could serve in the future to contrast PCa progression.

## Figures and Tables

**Figure 1 cancers-15-00948-f001:**
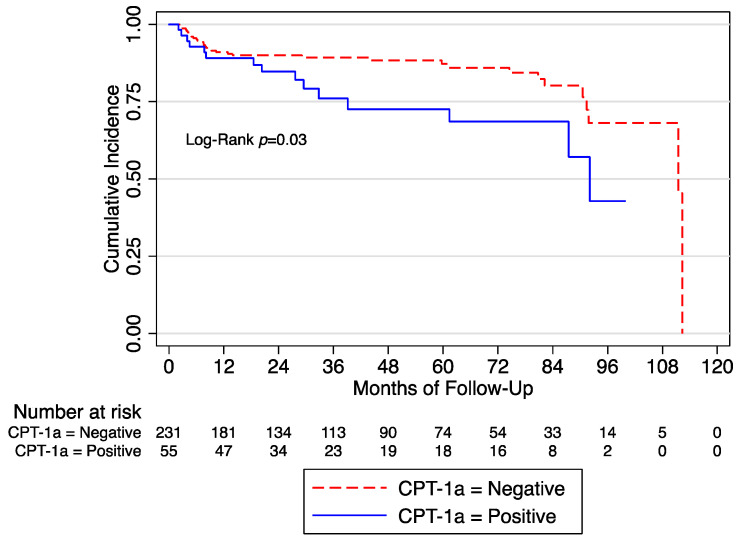
Biochemical recurrence free survival in the whole cohort according to the expression of CPT-1a. CPT = Anti-carnitine palmitoyl transferase.

**Table 1 cancers-15-00948-t001:** ATP citrate lyase expression according to IHC score.

	ATP Citrate Lyase		*p*-Value
	Negative (*n* = 203)	Positive (*n* = 187)	
Age (years), median (IQR)	71.0 (65.0–77.0)	68.0 (63.0–72.0)	<0.01
PSA (ng/mL), median IQR)	5.7 (2.11–8.9)	7.57 (5.6–11.5)	<0.01
Fasting glucose (mg/dL), median (IQR)	98.0 (88.0–111.0)	95.0 (87.0–108.5)	0.23
Total cholesterol (mg/dL), median (IQR)	183.0 (157.0–210.0	190.5 (159.0–216.0)	0.38
Triglycerides (mg/dL), median (IQR)	100.0 (65.0–150.0)	101.5 (73.0–136.0)	0.84
Diabetes, *n* (%)	28 (26.92)	44 (15.38)	<0.01
Group, *n* (%)			<0.01
BPH	86 (42.36)	18 (9.63)	
PC	117 (57.64)	169 (90.37)	
ISUP Gleason score, *n* (%)			0.21
1	35 (29.91)	47 (27.81)	
2	52 (44.44)	58 (34.32)	
3	22 (18.80)	44 (26.04)	
4	3 (2.56)	11 (6.51)	
5	5 (4.27)	9 (5.33)	
Pathological stage, *n* (%)			0.55
T2	84 (71.79)	113 (66.86)	
T3	21 (17.95)	32 (19.05)	
T4	12 (10.26)	24 (14.29)	
Classification risk of PC, *n* (%)			0.57
Low risk	42 (35.90)	58 (34.32)	
Intermediate risk	54 (46.15)	72 (42.60)	
High risk	21 (17.95)	39 (23.08)	
Ki-67 positive score, *n* (%)	20 (9.85)	33 (17.65)	0.02
AR positive score, *n* (%)	84 (41.38)	98 (52.41)	0.03
PSMA positive score, *n* (%)	58 (28.57)	90 (48.13)	<0.01
IR-α positive score, *n* (%)	105 (51.72)	154 (82.35)	<0.01
IR-β positive score, *n* (%)	9 (4.43)	14 (7.49)	0.20
IGF-1R positive score, *n* (%)	23 (11.33)	41 (21.93)	<0.01
SRSF-1 positive score, *n* (%)	80 (39.41)	108 (57.75)	<0.01
CPT1-a positive score, *n* (%)	30 (14.78)	35 (18.72)	0.30
SCD-1 positive score, *n* (%)	24 (11.82)	41 (21.93)	<0.01
SREBP1 positive score, *n* (%)	39 (19.21)	57 (30.48)	0.01
FAS positive score, *n* (%)	68 (33.50)	144 (77.01)	<0.01
ACC-1 positive score, *n* (%)	31 (15.27)	113 (60.43)	<0.01

IRS = immunoreactivity score BPH = Benign Prostatic Hyperplasia; PCa = prostate cancer; IQR = interquartile range; AR = Androgenic receptor; IR = insulin receptor; IGF-1R = insulin-like growth factor-1 receptor; PSMA = prostate specific membrane antigen; SRSF-1 = Serine/arginine-rich splicing factor 1; FAS = fatty acid synthase; CPT-1a = Carnitine palmitoyltransferase 1a; SCD-1 = Stearoyl-CoA desaturase-1; SREBP-1 = Sterol regulatory element-binding protein-1; AC-1 = Acetyl-CoA Carboxylase-1.

**Table 2 cancers-15-00948-t002:** Carnitine palmitoyltransferase-1a expression according to IHC score.

	Carnitine Palmitoyltransferase-1a		*p*-Value
	Low-IRS (*n* = 325)	High-IRS (*n* = 65)	
Age (years), median (IQR)	70.0 (64.0–74.0)	68.0 (64.0–74.0)	<0.01
PSA (ng/mL), median IQR)	6.43 (4.05–10.0)	7.0 (4.9–10.01)	<0.01
Fasting glucose (mg/dL), median (IQR)	96.0 (88.0–109)	99.0 (87.0–111.0)	0.71
Total cholesterol (mg/dL), median (IQR)	187.0 (158.0–214.0)	179.0 (152.0–200.0)	<0.01
Triglycerides (mg/dL), median (IQR)	99.0 (68.0–137.0)	121.0 (73.0–170.0)	<0.01
Diabetes, *n* (%)	61 (18.77)	11 (16.92)	0.73
Group, *n* (%)			0.02
BPH	94 (28.92)	10 (15.38)	
PC	231 (71.08)	55 (84.62)	
ISUP Gleason score, *n* (%)			0.84
1	68 (29.44)	14 (25.45)	
2	89 (38.53)	21 (38.18)	
3	52 (22.51)	14 (25.45)	
4	12 (5.19)	2 (3.64)	
5	10 (4.33)	4 (7.27)	
Pathological stage, *n* (%)			0.95
T2	159 (69.13)	37 (67.27)	
T3	42 (18.26)	11 (20.0)	
T4	29 (12.61	7 (12.73)	
Classification risk of PC, *n* (%)			0.14
Low risk	87 (37.6)	13 (23.64)	
Intermediate risk	98 (42.42)	28 (50.91)	
High risk	46 (19.91)	14 (25.45)	
Ki-67 positive score, *n* (%)	53 (15.73)	12 (22.64)	0.21
AR positive score, *n* (%)	25 (12.02)	40 (21.98)	<0.01
PSMA positive score, *n* (%)	36 (14.88)	29 (19.59)	0.22
IR-α positive score, *n* (%)	10 (7.63)	55 (21.24)	<0.01
IR-β positive score, *n* (%)	57 (15.53)	8 (34.78)	0.02
IGF-1R positive score, *n* (%)	53 (16.26)	12 (18.75)	0.62
SRSF-1 positive score, *n* (%)	23 (11.39)	42 (22.34)	<0.01
ATP-citrate lyase positive score, *n* (%)	30 (14.78)	35 (18.72)	0.29
SCD-1 positive score, *n* (%)	49 (15.08)	16 (24.62)	0.06
SREBP1 positive score, *n* (%)	41 (13.95)	24 (25.00)	0.01
FAS positive score, *n* (%)	15 (8.43)	50 (23.58)	<0.01
ACC-1 positive score, *n* (%)	29 (11.79)	36 (25.00)	<0.01

IRS = immunoreactivity score BPH= Benign Prostatic Hyperplasia; PCa = prostate cancer; IQR = interquartile range; AR = Androgenic receptor; IR = insulin receptor; IGF-1R = insulin-like growth factor-1 receptor; PSMA = prostate specific membrane antigen; SRSF-1 = Serine/arginine-rich splicing factor 1; FAS = fatty acid synthase; CPT-1a = Carnitine palmitoyltransferase 1a; SCD-1 = Stearoyl-CoA desaturase-1; SREBP-1 = Sterol regulatory element-binding protein-1; AC-1 = Acetyl-CoA Carboxylase-1.

**Table 3 cancers-15-00948-t003:** Univariate logistic regression between immunohistochemistry results and clinical and pathological variables in PC patients.

	ATPLy + vs. − (OR 95% CI)	CPT1a, + vs. − (OR 95% CI)	SCD + vs. − (OR 95% CI)	SREBP + vs. − (OR 95% CI)	FAS + vs. − (OR 95% CI)	AC-1 + vs. − (OR 95% CI)
**PSA, continuous**	1.01 (0.98–1.03)	1.00 (0.97–1.02)	0.98 (0.95–1.01)	0.96 (0.92–1.00)	1.00 (0.98–1.03)	0.99 (0.97–1.01)
**Fasting blood glucose, continuous**	0.99 (0.98–1.01)	1.00 (0.99–1.02)	0.99 (0.98–1.01)	1.00 (0.99–1.01)	0.99 (0.98–1.00)	0.99 (0.98–1.01)
**Total cholesterol, continuous**	0.99 (0.98–1.00)	0.99 (0.98–1.01)	0.99 (0.98–1.00)	0.99 (0.98–1.00)	0.99 (0.99–1.00)	0.99 (0.98–1.00)
**Triglycerides, continuous**	1.00 (0.99–1.01)	1.00 (0.99–1.01)	0.99 (0.98–1.00)	0.99 (0.98–1.00)	0.99 (0.99.1.00)	0.99 (0.98–1.00)
**Diabetes, yes vs. no**	1.11 (0.58–2.16)	0.82 (0.49–2.43)	1.58 (0.74–3.39)	0.51 (0.22–1.21)	0.50 (0.26–0.97)	1.60 (0.83–3.06)
**Pathological stage, pT3/4 vs. pT2**	1.27 (0.76–2.12)	1.08 (0.79–2.04)	0.94 (0.49–1.80)	1.04 (0.60–1.85)	0.71 (0.42–1.20)	1.30 (0.93–1.80)
**ISUP Gleason, ≥4 vs. <4**	1.82 (0.77–4.30)	0.75 (0.44–3.02)	1.53 (0.61–3.82)	0.79 (0.30–2.04)	1.47 (0.60–3.60)	1.21 (0.98–1.52)
**AR, + vs. −**	1.71 (1.06–2.77) ^†^	2.27 (1.24–4.16) ^†^	2.87 (1.53–5.39) ^†^	2.16 (1.25–3.73) ^†^	2.19 (1.30–3.69) ^†^	3.65 (2.22–5.93) ^†^
**PSMA, + vs. −**	1.12 (0.70–1.80)	0.97 (0.54–1.75)	1.16 (0.64–2.12)	0.94 (0.55–1.61)	1.64 (1.00–2.71) ^†^	1.80 (1.13–2.89) ^†^
**Ki-67, + vs. −**	1.33 (0.71–2.50)	1.37 (0.66–2.84)	2.16 (1.07–4.32) ^†^	1.02 (0.51–2.04)	1.67 (0.83–3.38)	1.11 (0.60–2.03)
**IR-α, + vs. −**	2.56 (1.43–4.56) ^†^	2.55 (1.03–6.27) ^†^	1.20 (0.56–2.56)	1.93 (0.92–4.05)	3.31 (1.84–5.95) ^†^	9.99 (4.35–22.93) ^†^
**IR-β, + vs. −**	1.08 (0.45–2.59)	2.45 (1.01–6.11) ^†^	1.24 (0.44–3.51)	1.33 (0.52–3.38)	1.77 (0.63–4.95)	1.85 (0.77–4.43)
**IGF-1R, + vs. −**	1.30 (0.73–2.32)	0.96 (0.47–1.95)	1.01 (0.49–2.07)	0.35 (0.16–0.78) ^†^	0.80 (0.44–1.44)	1.18 (0.67–2.05)
**ATPLy + vs. −**	-	1.26 (0.69–2.32)	1.43 (0.76–2.68)	1.41 (0.81–2.47)	4.84 (2.84–8.25) ^†^	4.97 (2.95–8.39) ^†^
**CPT1a, + vs. −**	1.26 (0.69–2.32)	-	2.15 (1.08–4.24) ^†^	2.95 (1.58–5.49) ^†^	2.16 (1.05–4.41) ^†^	2.12 (1.16–3.87) ^†^
**SCD + vs. −**	1.43 (0.76–2.68)	2.15 (1.08–4.24) ^†^	-	2.87 (1.53–5.39) ^†^	3.17 (1.42–7.04) ^†^	2.63 (1.40–4.91) ^†^
**SREBP + vs. −**	1.41 (0.81–2.47)	2.95 (1.57–5.48) ^†^	2.87 (1.53–5.39) ^†^	-	1.74 (0.94–3.21)	2.53 (1.45–4.40) ^†^
**FAS + vs. −**	4.84 (2.84–8.25) ^†^	2.16 (1.05–4.41) ^†^	3.17 (1.42–7.04) ^†^	1.74 (0.94–3.21)	-	11.29 (5.76–22.14) ^†^

OR = odds ratio; CI = confidence interval; PSA = prostate-specific antigen; ISUP = International Society of Urological. Pathology; AR = androgen receptor; PSMA = prostate specific antigen; IR = insulin receptor; IGF-1R = insulin growth factor-1 receptor; ATPLy = ATP lyase; CPT = Anti-carnitine palmitoyl transferase; SCD = Stearoyl-CoA desaturase-1; SREBP = Sterol regulatory element-binding protein; FAS = fatty acid synthase; AC-1 = Acetyl-CoA Carboxylase-1; ^†^
*p* < 0.05.

## Data Availability

The data presented in this study are available on request from the corresponding author.

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
