# Peer review of "Is There a Role of Warburg Effect in Prostate Cancer Aggressiveness? Analysis of Expression of Enzymes of Lipidic Metabolism by Immunohistochemistry in Prostate Cancer Patients (DIAMOND Study)"

_cancers, 2023, doi:10.3390/cancers15030948_

Round 1

Reviewer 1 Report

Authors described the Warburg effect is associated with prostate cancer progression. The results were confirmed by several immunohistochemical staining. It is highly interesting for readers. However, the structure of reference and tables should be refined. Some spell errors should be corrected (line 25). Besides, I wonder how to define the expression of antibodies by the percentage of score (line 122-123)? 

Author Response

Comments:

Authors described the Warburg effect is associated with prostate cancer progression. The results were confirmed by several immunohistochemical staining. It is highly interesting for readers. However, the structure of reference and tables should be refined. Some spell errors should be corrected (line 25). Besides, I wonder how to define the expression of antibodies by the percentage of score (line 122-123)? 

Response:

We would like to thank the reviewer for his time dedicated and his precious comments.

  1. We have revised reference style according to the journal instructions. We have also refined tables in particular the number 4.
  2. We have revised the whole manuscript.
  3. As concerning expression of AB we have applied the following methods:

“Immunohistochemical slides were evaluated by three pathologists (G.B., E.P. and R.C.) with no information on patient clinical data. As previously described, a pathologist marked all sections with hematoxylin and eosin considering PCa tissue with the highest grade. The scoring system included a combined analysis of staining intensity (IS) and percentage of immunoreactive cells (extent score; ES), as previously described 18,19.

Intensity of staining (IS) was graded on a 0–3 scale (0 = absent staining, 1 = weak staining, 2 = moderate staining, 3 = strong staining). Five categories (0–4) of percentage of SRSF1 immunopositive cells (Extent Score [ES]) were identified: <5%; 5–30%; 31–50%; 51–75%; >75%. IS was multiplied by ES to obtain the immunoreactivity score (IRS); low (L‐IRS) and high (H‐IRS) expression of SRSF1 were defined as IRS <6 and IRS ≥6, respectively.”

This method is reliable and it is generally used in this type of study.

Reviewer 2 Report

In this paper, Russo et al. analyse the expression of enzymes of lipidic metabolism by immunohistochemistry in prostate cancer patients.

General comment: The results of a paper are relevant.

Specific points:

Table 4: Univariate analysis is missing the p-values. Are the results marked with a sign statistically significant? The authors should make this clear.

Table 4 is listing the results of the univariate analysis. However, it is not clear where are the results of a multivariate analysis.

The sum of the numbers of patients in Table 2 under the Pathological stage is 168 and not 169. The authors should state somewhere where this difference is coming from (e.g. unknown stage for one patient or something else).

Author Response

We would like to thank the reviewer for his time dedicated and his precious comments.

In this paper, Russo et al. analyse the expression of enzymes of lipidic metabolism by immunohistochemistry in prostate cancer patients.

General comment: The results of a paper are relevant.

Specific points:

Table 4: Univariate analysis is missing the p-values. Are the results marked with a sign statistically significant? The authors should make this clear.

Response:

We agree with the Reviewer. We have added a part in the legend reporting “† p<0.05”.

Table 4 is listing the results of the univariate analysis. However, it is not clear where are the results of a multivariate analysis.

Response:

We did not perform a multivariate logistic analysis in the original article and we are sorry for the mistake. Indeed, we updated the manuscript by reporting this new analysis. We have also added few new lines in the discussion.

The sum of the numbers of patients in Table 2 under the Pathological stage is 168 and not 169. The authors should state somewhere where this difference is coming from (e.g. unknown stage for one patient or something else).

Response:

We would like to thank you for this underlined mistake. It was only a typo in reporting results. The patient was a pT2. We have corrected the table.